# Production of Refractory Materials Using a Renewable Source of Silicon Dioxide

Abdurassul Zharmenov [1,†], Svetlana Yefremova [1,*,†], Baimakhan Satbaev [2,†], Nurgali Shalabaev [2], Serik Satbaev [2], Sergey Yermishin [1] and Askhat Kablanbekov [1]

[1] National Center on Complex Processing of Mineral Raw Materials of the Republic of Kazakhstan RSE, Almaty 050036, Kazakhstan

[2] RSE Astana Branch of the National Center on Complex Processing of Mineral Raw Materials of the Republic of Kazakhstan, Nur-Sultan 010000, Kazakhstan

\* Correspondence: s_yefremova@cmrp.kz or secretar_rgp@mail.ru; Tel.: +7-777-587-9192

† These authors contributed equally to this work.

**Abstract:** Organization of environmentally-friendly production of refractory materials based on the principles of cost-effective use of energy and material resources through use of energy-saving technologies and replacement of natural raw materials with industrial and agricultural waste is gaining relevance. Scientists are increasingly interested in creating high-temperature materials using silica of plant origin. Its source is rice husk, a multi-tonnage waste from rice production. Organo-mineral in its nature, rice husk determines the uniqueness of the structure and properties of the materials obtained from it. Use of this waste allows us to produce porous, high-strength silicon carbide refractories with properties corresponding to classical analogs, while benefiting from environmental, economic and technological aspects. The lack of industrial production of refractories using rice husk ash, despite the positive results of scientific research, indicates insufficient study of the issue with certain gaps in this area. This review is intended to help researchers to identify existing problems and outline further actions necessary to ensure that the scientific results are implemented in production.

**Keywords:** rice husk; agricultural silicon-containing waste; silica; high-temperature materials; porous refractories; high-strength refractories; silicon carbide-containing refractories; hybrid materials

## 1. Introduction

Effective protection of equipment and technical structures working in high-temperature areas from the damaging effects of aggressive environments has always been a pressing issue for the most important sectors of industrial economies: metallurgical, chemical, petrochemical and others. The feasibility of creating new high-temperature materials to meet the needs of these industries is driven by the increasing demands of consumers and the need to improve the operating conditions and reduce manufacturing energy costs. The modern trend of the development of high-temperature materials' production on the global level is aimed at developing resource-saving composites of a new generation with high fire and heat resistance, including in chemically aggressive environments, while being environmentally friendly and providing improved quality with the final products [1]. Some industries achieve resource conservation using industrial waste as a raw material resource. For example, metallurgical waste is used to create high-quality building materials [2–8]. The value of its use has also been proven for the synthesis of high-temperature materials [9–11].

In recent years, scientists' interest in creating hybrid (organo-mineral) composite materials has increased [12]. The idea of introducing silica ash of rice husk, along with the traditional ingredients, in the composition of high-temperature ceramics and bricks was proposed [13]. Such work has recently been carried out in many countries (Thailand, Vietnam, Malaysia, China, South Korea, South Africa, Brazil). Rice husk is a large-scale

biogenic residue within the global economy. Rice is the second most important and common food product. The world production of rice amounts to 800 million tons. Rice husk accounts for about 20–25% wt. of this mass. Known methods of processing this multi-tonnage waste produce even more hazardous waste. The most common process is incineration for energy [14]. However, this produces toxic gases and ash. The formation of the latter is due to the presence of silica in rice husk amounting to ~20% wt. The $SiO_2$ content in the ash, depending on the feedstock and its combustion conditions, varies within the range of 80–95% wt. [15,16]. This is called biogenic silica. Traditionally, porous silica has been produced on an industrial scale using water glass at high (>1400 °C) temperatures. Yet, it is not only an expensive but also an environmentally harmful process. Each ton of silica produces 0.23 tons of $CO_2$, 0.74 tons of $Na_2SO_4$ and 20 tons of wastewater [17]. In this regard, obtaining silica using eco-friendly methods such as biomass recycling is very attractive. Biogenic silica to be used in advanced technologies must not contain carbon and must have a high degree of purity. In addition, it should be characterized by a developed system of pores and have a significant specific surface area. Moreover, its structure should be amorphous. Three main approaches to obtaining biogenic silica from rice husk with the above properties have been proposed in the literature [17] with appropriate references. First of all, is the combusting of rice husk to produce ash; secondly, there is pre-treatment of raw materials; finally, some opt for the post-treatment of rice husk ash.

In the combustion process, the fundamental factors are the temperature and residence time of the raw material. Increasing these parameters increases the degree of purity of the silica produced. However, its degree of crystallization increases, and the specific surface area and total pore volume decrease. The purity of biogenic silica is also affected by the heating rate. The higher it is, the greater the amount of unburned carbon in silica, and 600 °C is typically recommended as the optimum combustion temperature. At this temperature, silica reaches 97.2% wt. purity, whereas the specific surface area is 220 $m^2 g^{-1}$ and the pore volume is 0.26 $cm^3 g^{-1}$.

Pre-treatment of rice husk before combustion is carried out to improve these indicators. For example, washing with water to remove alkali metals or leaching with acid to remove alkaline-earth elements. The main thing is to exclude the presence of ash-forming elements. Accordingly, after combustion of pre-treated rice husk, its degree of purity increases to 99.8% wt., the specific surface area increases to 353 $m^2 g^{-1}$ and the volume of pores increases to 0.52 $cm^3 g^{-1}$. The presence of alkali metals influences the degree of crystallization of silica. If the crystallization of silica in the combustion of untreated rice husk occurs already at 600–900 °C, after the pretreatment of rice husk, $SiO_2$ remains amorphous at higher combustion temperatures (~1000 °C).

To achieve the desired performance of biogenic silica, it is possible to use a combination of pre-treatment methods before combustion, select the combustion conditions and conduct the final treatment of the resulting product. For example, in [18], a series of methods involving treatment of rice husk by hexadecyltrimethylammonium hydroxide solution in an autoclave produced biogenic silica with a specific surface area of 1210 $m^2 g^{-1}$ and volume of pores of 1.0 $cm^3 g^{-1}$. However, it should be taken into consideration that all these methods increase the cost of silica and are not always eco-friendly. In each case, it is necessary to choose the most appropriate method of silica production. Its purity and characteristics will be determined by the objectives of the further application of the material.

Fine silica, resulting from rice husk burned for energy production, has a low bulk weight, easily gets into the air and has a negative impact on human health. This makes the environmental situation extremely difficult. Many researchers have proposed the use of silicon dioxide ash from rice husk to produce pure silicon and its compounds, concrete, cement and refractory ceramics [13,14,17,19–37]. It has been shown that rice husk ash, due to its properties (primarily the presence of silicon dioxide in a nanostructured amorphous form with high purity and low thermal conductivity), is a promising source of silicon for the production of ceramics [36], including insulating and high-strength-type (mullite, forsterite, cordierite, carbide-silicon) refractories [14].

In general, there is a large variety of refractory materials, which differ in their properties and so are used differently. Refractories are classified according to a number of features. Depending on the physical and chemical properties of raw materials, they are divided into acidic (silica, aluminosilicate, zirconia), basic (magnesite, dolomite, magnesia-chrome) and neutral (carbon graphite, chromite, alumina) refractories. According to the melting temperature, a distinction is made between normal (1580–1780 °C), high (1780–2000 °C) and super (over 2000 °C) refractories. According to the method of production, they are divided into molded and unmolded, flame retardant and thermally conductive [33].

Kazakhstan, as a country with developed metallurgical, chemical and other industrial sectors, is in need of different types of high-temperature materials. Unfortunately, refractories are fully imported from abroad, although all the necessary raw materials, both natural and anthropogenic, including rice husk, are available in sufficient quantities in the country for production. Despite the great variety of new high-temperature materials with different physical and chemical properties proposed by Kazakh scientists [9–11] and the possibility of their widespread use in the above-mentioned industries, the task of organizing the local production of these materials remains unresolved, a problem existing not only in our country. Plus, positive results in the field of studying the possibility of using rice husk ash in the production of refractory ceramics, on a global scale, have not been achieved since 1913 [37]. There are a number of unresolved issues. The main one is detachment from production. All the results have a purely academic orientation. There is a lack of any significant contribution to the refractory industry with the application of new refractories in practice [36,37]. According to the experts [37], solving this problem will produce a real boom in the industry of high-temperature materials, including refractories.

In this regard, the purpose of this review is to study the current state of the art in the field of creating refractory ceramics using rice husk ash as a renewable source of silica. The work presents recent advances in rice husk ash application for the synthesis of porous and high-strength types of refractory materials. This review will be useful for the creation of sustainable refractories with desired properties when replacing the traditional raw materials (quartz, refractory clay) with rice husk ash, used alone or in combination with other production waste.

## 2. Porous Refractory

### 2.1. Diatomite, Rice Husk Ash, Sawdust

Thang [33] obtained refractory composites using the following available raw materials: low-quality diatomite (DE, 90 μm), rice husk ash (RHA, 90 μm), sawdust (SD, 250 μm). The ingredients needed to prepare the charge are shown in Table 1. In the process, 40% wt. of water was added to the mixture, the samples were pressed at 10 MPa and then they were dried for 72 h at 30 °C and then dried for 24 h at 60 °C. The samples were finally heated to 1200 °C for 2 h, at a rate of 1 °C min$^{-1}$ while heating from room temperature to 600 °C and then 2 °C min$^{-1}$ afterward. The cooling process was carried out at a rate of 3 °C min$^{-1}$.

The use of RHA made it possible to replace diatomite in equal proportions. The use of highly porous raw materials, namely DE and RHA, according to the author, ensured the production of lightweight refractory composites with high porosity and low thermal conductivity. The presence of sawdust as a burning additive affects the index of porosity. The porosity increases with increasing amounts of sawdust. At the same time, mechanical strength (in bending and compression) and thermal conductivity decrease. The maximum sawdust content did not exceed 35% wt. This is because when increasing its quantity up to 40% wt. and more, the mixture lost plasticity and it was impossible to mold the bricks in which form the refractory samples were to be obtained (220 × 110 × 65 mm).

Low thermal conductivity of refractories ensures effective protection of the furnace surface during the combustion process. The experimental samples were characterized by high stability (up to 1400 °C): they kept their shape when heated up to 1110 °C and collapsed at temperatures above 1413 °C. Such characteristics of new refractory composites confirm

their compliance with the requirements of modern standards and allow the obtained materials to be used in harsh conditions, namely at high ambient temperatures.

**Table 1.** Composition and properties of porous DE/RHA/SD-based refractory. Data from [33].

| Samples | Raw Materials, % wt. | | | Engineering Properties | | Mechanical Properties | | Thermal Properties | |
|---|---|---|---|---|---|---|---|---|---|
| | DE | RHA | SD | Volumetric Weight, g cm$^{-3}$ | Porosity, % | Bending Strength, MPa | Compressive Strength, MPa | Thermal Shock Resistance at 1000 °C, Time | Thermal Conductivity, W (m K)$^{-1}$ |
| M0 | 50.0 | 50.0 | 0 | 0.66 | 75.32 | 1.78 | 17.35 | 78 | 0.1035 |
| M5 | 47.5 | 47.5 | 5.0 | 0.61 | 74.28 | 1.75 | 17.24 | 75 | 0.1024 |
| M10 | 45.0 | 45.0 | 10.0 | 0.57 | 78.13 | 1.72 | 17.03 | 72 | 0.1003 |
| M15 | 42.5 | 42.5 | 15.0 | 0.53 | 81.64 | 1.72 | 16.84 | 72 | 0.0984 |
| M20 | 40.0 | 40.0 | 20.0 | 0.48 | 83.08 | 1.70 | 16.68 | 70 | 0.0968 |
| M25 | 37.5 | 37.5 | 25.0 | 0.44 | 86.76 | 1.68 | 16.41 | 68 | 0.0941 |
| M30 | 35.0 | 35.0 | 30.0 | 0.41 | 88.51 | 1.66 | 16.11 | 66 | 0.0911 |
| M35 | 32.5 | 32.5 | 35.0 | 0.37 | 92.45 | 1.61 | 15.78 | 61 | 0.0878 |

This study showed that the utilization of agricultural and forest industry waste, namely RHA and SD, is useful not only in terms of economic and environmental benefits. With their use as raw materials, lightweight porous refractories were obtained. The author plans to study the microstructure of these innovative materials in the future, which will allow for a deeper understanding of the processes under study. At this stage, however, what is important is not just the fact that waste can be used for production of this or that product but also the fact of obtaining new high-quality refractory materials using waste.

### 2.2. Refractory Clay, Rice Husk Ash, Wollastonite Microfibers

The effect of porosity on the performance properties of refractories was also noted by Silva et al. [34]. They studied the replacement of a part of the refractory clay (C, % wt.: $SiO_2$—23.19; $Al_2O_3$—54.40; $Fe_2O_3$—4.98; other oxides—17.43) with rice husk ash (RHA, % wt.: $SiO_2$—89.06; other oxides—10.94) and its combination with wollastonite microfibers (W, % wt.: $SiO_2$—43.25; CaO—55.19; other oxides—1.59) (Table 2). X-ray diffraction (XRD) showed that the clay was mainly formed by crystalline materials with a predominance of the aluminum oxide phase. Judging by the halo in the region of $2\theta = 20$–$30°$, amorphous material is also present in the clay. According to XRD data, RHA is mainly formed by amorphous material represented by a halo with a maximum at $2\theta = 22.5°$. W, on the other hand, is represented by calcium oxide and silicon dioxide in crystalline form.

To obtain the experimental samples, the mixtures were processed in a planetary mill in two stages of 3 min each. In the second stage, 10% wt. of water was added. The samples were pressed using a uniaxial press at 35 MPa with a size of $150 \times 30 \times 20$ mm. Next, the samples were dried at 60, 80 and 105 °C to a constant weight (held for 24 h at the last temperature) and sintered at 1000 °C. Cooling was performed under natural conditions.

It has been shown that the formation of pores in refractory composites is associated with two factors, firstly, with the mullitization reaction. At the sintering stage, there is mutual diffusion of aluminum oxide and silicon oxide ions with the formation of aluminosilicate liquid. With an increasing temperature, the liquid is gradually enriched with aluminum ions until their number reaches a stoichiometric amount in terms of mullite. With an excess of aluminum oxide, a smaller stretch of the liquid phase is formed, and accordingly, an excess of silicon oxide is eliminated. In the case of an excess of silica, its part, not involved in the formation of mullite, turns into cristobalite and fills the available pore space, contributing to a reduction in porosity. This aspect occurs not only when replacing

the refractory clay with silica-containing raw materials, in particular, rice husk ash, but also when using clays with a high $SiO_2/Al_2O_3$ ratio.

**Table 2.** Composition and properties of A/S/W-based refractory. Data from [34].

| Samples | Raw Materials, % wt. | | | Engineering Properties, % | | | Mechanical Properties, % | | Thermal Properties | | | |
|---|---|---|---|---|---|---|---|---|---|---|---|---|
| | C | RHA | W | Water Absorption | Apparent Porosity | Bulk Density | Linear Retraction after the Sintering Process | Mass Variation | Maximum Number of Cycles for Thermal Shock Fracture/ Thermal Shock Resistance at Different Temperatures | | | Thermal Conductivity, W (m K)$^{-1}$ |
| | | | | | | | | | 500 °C | 850 °C | 1000 °C | |
| A | 100 | 0 | 0 | 7.82 | 12.99 | 1.66 | 7.78 | 4.73 | 4/0.3 | 2/0.5 | 1/1.0 | 0.1659 |
| AS10 | 90 | 10 | 0 | 10.23 | 15.76 | 1.54 | 6.36 | 4.63 | 2/0.84 | 1/1.0 | 1/1.0 | 0.1623 |
| AS10W5 | 85 | 10 | 5 | 13.04 | 19.19 | 1.47 | 3.94 | 3.88 | 3/0.82 | 2/0.65 | 1/1.0 | 0.1559 |
| AS10W10 | 80 | 10 | 10 | 14.43 | 22.94 | 1.59 | 2.6 | 3.05 | 2/0.83 | 1/1.0 | 1/1.0 | 0.1523 |
| AS10W20 | 70 | 10 | 20 | 16.60 | 25.45 | 1.53 | 1.64 | 2.55 | 1/1.0 | 1/1.0 | 1/1.0 | 0.1446 |

Another factor is the granulometric composition. An increase in porosity in refractories is observed when using raw materials that have different grain sizes. This is achieved, for example, by replacing fine material with a coarser ingredient, which in this study was wollastonite. Table 2 shows that introduction of wollastonite significantly increased the porosity of the obtained refractories, although the authors proposed that a difference in grain size of the ingredients used to prepare refractories will contribute to the growth of porosity even without introduction of microfibers. The authors explained the high porosity of the synthesized refractories by the incomplete mullitization reaction in addition to the presence of a granulometric aspect. In general, introduction of wollastonite somewhat worsens the mechanical properties of refractory composites, which affects their thermal properties. Based on the maximum number of cycles for thermal shock fracture and thermal shock resistance at 500, 850, and 1000 °C, it can be concluded that the refractories of this composition can be operated at temperatures up to 1000 °C.

In general, the authors showed that rice husk ash, due to its high content of silica, mainly in the amorphous form, represents an alternative source of silica for the production of ceramics. When mixed with wollastonite, it provides an increase in porosity and water absorption, which is associated with the incomplete mullitization process and the granulometric aspect.

### 2.3. Kaolin, Rice Husk Silica, Steel Fibers

Stochero et al. [35] observed a similar pattern of porosity changes to that described above when kaolin (KC, % wt.: $SiO_2$—57.83; CaO—0.13; MgO—0.36; $Fe_2O_3$—2.25; $Al_2O_3$—27.52; $Na_2O$—<0.001; $K_2O$—1.87; $TiO_2$—0.38; MnO—<0.01; loss to fire—8.63) was replaced with silica extracted from rice husk (RHS, % wt.: $SiO_2$—91.48; CaO—0.36; MgO—0.32; $Fe_2O_3$—0.05; $Al_2O_3$—ND; $Na_2O$—0.04; $K_2O$—1.40; $TiO_2$—0.003; MnO—0.32; $SO_3$—0.15; $P_2O_5$—0.45; loss to fire—3.50) and steel fibers (SF) were added. RHS was obtained by burning rice husk in a fluidized bed at 780 °C for 70 s. According to XRD data, the silica had an amorphous-crystalline structure. Judging by the peaks at 21.89° and 36.02°, it could be attributed to quartz with admixtures of cristobalite.

The samples were pressed at 20 MPa with a size of 150 × 30 × 20 mm. The moisture content was 12% wt. The samples were dried at room temperature for 20 days, then in

an oven at $105 \pm 10$ °C for 24 h. The samples were then heated in three stages: at a rate of 5 °C min$^{-1}$ they were heated to 150 °C and held there for 10 min; then, at a rate of 3 °C min$^{-1}$, they were heated to 500 °C and held there for 10 min; and in the last stage, at a rate of 5 °C min$^{-1}$, they were heated to 1300 °C and held there for 30 min. The samples were cooled naturally to room temperature.

Replacement of kaolin with RHS alone contributed to a decrease in porosity and, accordingly, an increase in bulk density due to improved packaging of the ingredients in the refractory ceramics' matrix. With the introduction of steel fibers, as a result of their random distribution, the packing order was disturbed, and the porosity increased as their amount was increased (Table 3). In Table 3, it can be seen that the introduction of RHS increases the tensile strength in bending up to 28.34 MPa, while the subsequent introduction of SF significantly reduces this parameter. However, the fiber-containing refractory samples are more ductile and are characterized by higher thermal shock-resistance values. The authors [35] attributed some deterioration of this parameter for the sample K20S9F to the high porosity index. In general, replacement of a part of kaolin with silica from rice husk and steel fibers opens up the possibility of producing refractory ceramics with improved mechanical and thermal properties.

**Table 3.** Composition and properties of KC/RHS/SF-based refractory. Data from [35].

| Samples | Raw Materials, % Vol. | | | Engineering Properties | | Tensile Strength in Bending, MPa | Thermal Shock Resistance at 1200 °C |
|---|---|---|---|---|---|---|---|
| | KC | RHS | SF | Apparent Bulk Density, g cm$^{-3}$ | Porosity, % | | |
| KC | 100 | 0 | 0 | 2.15 | 11.5 | 24.41 | 1 |
| K20S | 80 | 20 | 0 | 2.25 | 8.8 | 28.34 | 1 |
| K20S3F | 77 | 20 | 3 | 2.35 | 10.5 | 14.00 | 0.25 |
| K20S6F | 74 | 20 | 6 | 2.45 | 12.5 | 13.75 | 0.27 |
| K20S9F | 71 | 20 | 9 | 2.60 | 13.0 | 13.00 | 0.36 |

*2.4. Kaolin, Rice Husk Silica*

In [13], refractory ceramics were obtained by replacing part of kaolin (KC, % wt.: $SiO_2$—57.83; CaO—0.13; MgO—0.36; $Fe_2O_3$—2.25; $Al_2O_3$—27.52; $Na_2O$—<0.001; $K_2O$—1.87; $TiO_2$—0.38; MnO—<0.01; loss to fire—8.63) with rice husk silica (RHS, % wt.: $SiO_2$—91.48; CaO—0.36; MgO—0.32; $Fe_2O_3$—0.05; $Al_2O_3$—ND; $Na_2O$—0.04; $K_2O$—1.40; $TiO_2$—0.003; MnO—0.32; $SO_3$—0.15; $P_2O_5$—0.45; loss to fire—3.50) without the addition of other ingredients. RHS was obtained by burning rice husks in a fluidized bed at temperatures below 650 °C.

Experimental samples sized $10 \times 3.13 \times 2.14$ (cm) were obtained by extrusion using a pressure of 28 kg cm$^{-3}$. The samples were dried at room temperature for 20 days, then in an oven at $105 \pm 10$ °C for 24 h. All samples were sintered at 1300 °C in a muffle furnace and cooled naturally when the oven was off.

Using XRD, it was found that rice husk silica contributes to the formation of cristobalite, without entering into the mullite formation reaction. In spite of the lower specific density (2.03 g cm$^{-3}$) compared to kaolin (2.73 g cm$^{-3}$), the bulk density of the refractories remained practically unchanged with the increasing amount of RHS in the mixture and the porosity decreased significantly (Table 4). This fact is explained by the more efficient packing of the ingredient grains in the mixture, which contributes to improved mechanical properties (tensile strength in bending and compressive strength increase, especially in samples KS10 and KS20). However, not all samples could pass the thermal shock-resistance tests. Based on a comprehensive analysis of thermomechanical properties, sample KS10, which had the most optimal composition and was characterized as having performed the best, could be recommended for production and application.

**Table 4.** Composition and properties of KC/RHS-based refractory. Data from [13].

| Samples | Raw Materials, % Vol. | | Engineering Properties | | Tensile Strength in Bending, MPa | Compressive Strength, MPa | Thermal Shock Resistance, at 1200 °C |
| | KC | RHS | Apparent Bulk Density, g cm$^{-3}$ | Porosity, % | | | |
|---|---|---|---|---|---|---|---|
| KC | 100 | 0 | 2.25 | 2.6 | 19.26 | 116.93 | 0.46 |
| KS5 | 95 | 5 | 2.25 | 2.1 | 18.75 | 115.0 | 0.47 |
| KS10 | 90 | 10 | 2.25 | 1.0 | 24.13 | 129.25 | 0.47 |
| KS20 | 80 | 20 | 2.25 | 0.37 | 27.98 | 140.06 | 0.73 |

*2.5. Rice Husk Ash with Additives Compared to Diatomite Silica*

The authors of an earlier work [15], while not denying the influence of porosity on the thermal conductivity of ceramic thermal insulators, attached more importance to the ordering of the crystal structure. They fabricated thermal insulators based on rice husk ash obtained by burning rice husk at 600–700 °C (RHA E, % wt.: $SiO_2$—83.68; $Al_2O_3$—0.17; $P_2O_5$—1.34; $K_2O$—3.65; CaO—1.03; MnO—0.66; $Fe_2O_3$—0.17; Cl—0.14; Br—0.09; C—9.07) mixed with wood sawdust (SD) by extrusion (Ex) and pressing (Pr) methods. RHA had the following physical properties: natural humidity—3.3%; theoretical density—2.2 g cm$^{-3}$; loss of ignition—10.6%. The main particle size was 0.21 mm. According to XRD data, RHA had an amorphous structure and was represented by the cristobalite phase. Depending on the method of production, different additives and binders were used (Table 5): polysaccharide (PS1), polyvinyl alcohol (PVA), bentonite (B).

The pastes were mixed in a mechanical mixer for 15 min. Samples of 10 × 20 × 60 mm were obtained for the tests. To determine the mechanical strength, cylinders with a diameter and height of 25 mm were obtained. RHA Ex was fired at 1350 °C, while RHA Pr was fired at 1250 °C. The heating rate was 250 °C h$^{-1}$. Samples of both types were cooled to room temperature.

**Table 5.** Composition and properties of RHA Ex and RHA Pr thermal insulators. Data from [15].

| Samples | Raw Materials, % wt. | | | | | | Porosity, % | Apparent Density, kg m$^{-3}$ | Compressive Strength, KPa | Thermal Conductivity, W (m K)$^{-1}$ | | | | | |
| | RHA E | PS1 | PVA | SD | B | H$_2$O | | | | 300 °C | 400 °C | 500 °C | 600 °C | 700 °C | 800 °C |
|---|---|---|---|---|---|---|---|---|---|---|---|---|---|---|---|
| Ex1 | 82 | 3 | - | 15 | - | 75 | 75 | 360 | 150 | 0.19 * | 0.2 * | 0.21 * | 0.22 * | 0.23 * | 0.24 * |
| Ex2 | 85 | 3 | - | 12 | - | 60 | 65 | 490 | 500 | - | - | - | - | - | - |
| Ex3 | 87 | 3 | - | 10 | - | 50 | 60 | 550 | 890 | - | - | - | - | - | - |
| Ex4 | 90 | 3 | - | 7 | - | 50 | 54 | 640 | 1050 | - | - | - | - | - | - |
| Ex5 | 92 | 3 | - | 5 | - | 50 | 50 | 700 | 1200 | - | - | - | - | - | - |
| Pr1 | 65 | - | 10 | 15 | 10 | 30 | 75 | 460 | 390 | 0.15 | 0.15 | 0.17 | 0.21 | 0.25 | 0.27 |
| Pr2 | 68 | - | 10 | 12 | 10 | 30 | 68 | 520 | 420 | - | - | - | - | - | - |
| Pr3 | 71 | - | 10 | 9 | 10 | 25 | 62 | 680 | 600 | - | - | - | - | - | - |
| Pr4 | 74 | - | 10 | 6 | 10 | 25 | 56 | 730 | 800 | - | - | - | - | - | - |
| Pr5 | 77 | - | 10 | 3 | 10 | 25 | 50 | 850 | 1000 | - | - | - | - | - | - |
| DS | - | - | - | - | - | - | 73 | 392 | 785 | 0.08 | 0.09 | 0.11 | 0.13 | 0.15 | 0.16 |

* The data are given for the RHA Ex sample with the following characteristics: porosity—70%, apparent density—660 kg m$^{-3}$, compression strength—647 KPa.

Comparison of the thermal properties of samples RHA Ex, RHA Pr and industrial diatomaceous silica (DS), which have similar physical and mechanical characteristics (Table 5), depending on the microstructure showed that the higher the degree of structure disorder, the lower the thermal conductivity of thermal insulators. Gonçalves et al. [15] explained this fact by the mechanism of heat transfer in ceramic materials based on crystalline lattice vibrations. The degree of structure disorder is determined by crystallite size, grain size and the presence of amorphous phases and voids. Thus, the presence of a vitreous phase in RHA Pr thermal insulators explains their lower thermal conductivity compared to RHA Ex thermal insulators. The DS microstructure is characterized by the presence of fine dispersed silicon particles and a large number of small pores randomly scattered on the surface. As a material with a very disordered structure, DS has better thermal conductivity values. Despite the higher thermal conductivity values of RHA Ex and RHA Pr samples relative to DS, in general, they are small and, along with high porosity and low density, allow us to use rice husk ash as a feedstock for the production of thermal insulators.

### 2.6. Ground and Unground Rice Husk Ash, Rice Husk Sol, Sodium Hexametaphosphate

Considering that rice husk ash or silica, isolated from their composition, are amorphous lightweight materials and contribute to the formation of porous refractories, Hossain et al. [31] tested the possibility of producing insulating bricks using them. As constituents, they used ground and unground rice husk ash (aggregates), isolated by alkaline extraction sol (binder), and added in small amounts of sodium hexametaphosphate (SHMP) (Table 6). SHMP and sol were stirred for 5 min. Then, aggregates were added and stirred for another 10 min. After adding a small amount of water, the mixture was poured into a mold that was held for 2 min on a vibrator, dried at 100 °C for 24 h and then sintered at 900, 1000, 1100 and 1200 °C for 2 h (one sample was only dried at 80 °C for 24 h, for the sake of comparison). X-ray phase analysis recorded formation of cristobalite at 900 °C and the tridymite crystal phase at 1200 °C. According to the electron microscopy, the samples obtained at 900–1100 °C had the same morphology (foamy structure, containing pores with sizes of 1–60 μm, which were irregular and interconnected). The sample obtained at 1200 °C was denser, with fewer pores. At high temperatures, silica particles were observed to enlarge as a result of diffusion and association of sol nanoparticles with silica particles of rice husk ash. After determining the main characteristics (porosity, strength and thermal conductivity) of the obtained samples and comparing them with literature data and industrial samples of insulating bricks (Table 6), it was shown that the new material obtained at 1000 °C meets the necessary requirements and can be used as an insulating refractory in various furnaces. At higher sintering temperatures, the degree of structural ordering increases due to the coarsening of silica particles and the appearance of crystalline phases, which, as shown in a previous paper [15], negatively affect the thermal properties of ceramic materials.

**Table 6.** Composition and properties of insulation refractory bricks. Data from [31].

| Samples | RHA, % wt. | | Sol, Dry Sol% wt. | SHMP, % wt. | Water, % wt. | | SiO$_2$, % | Apparent Bulk Density, g cm$^{-3}$ | Apparent Porosity, % | Cold Compressive Strength, MPa | Thermal Conductivity, W (m K)$^{-1}$ |
|---|---|---|---|---|---|---|---|---|---|---|---|
| | Unground, >200 μm | Ground, <200 μm | | | Add | From Sol | | | | | |
| s-1, green, 80 °C | 50 | 47.3 | 2.5 | 0.2 | 5 | 5.8 | - | - | - | - | - |
| s-2, 900 °C | 50 | 44.8 | 5 | 0.2 | 2 | 11.6 | - | 0.73 | 69.0 | 4.1 | 0.124 |

**Table 6.** *Cont.*

| Samples | RHA, % wt. | | | SHMP, % wt. | Water, % wt. | | SiO₂, % | Apparent Bulk Density, g cm⁻³ | Apparent Porosity, % | Cold Compressive Strength, MPa | Thermal Conductivity, W (m K)⁻¹ |
|---|---|---|---|---|---|---|---|---|---|---|---|
| | Unground, >200 µm | Ground, <200 µm | Sol, Dry Sol% wt. | | Add | From Sol | | | | | |
| s-3, 1000 °C | 50 | 42.30 | 7.5 | 0.2 | - | 17.5 | - | 0.76 | 67.0 | 5.0 | 0.132 |
| s-4, 1100 °C | 50 | 39.80 | 10 | 0.2 | - | 23.3 | 95 | 0.79 | 64.5 | 5.4 | 0.135 at 30 °C |
| s-5, 1200 °C | 50 | 37.30 | 12.5 | 0.2 | - | 29.2 | - | 0.90 | 59.0 | 6.5 | 0.146 |
| QG-0.8 * | | | | | | | ≥88 | 0.8 | ≥55 | 1.8 | - |
| HCR SI0.8 ** | - | | | | | | ≥88 | ≤0.85 | - | ≥1 | 0.55 at 350 °C |

* [38], cited from [31]; ** [39], cited from [31].

## 2.7. Ghanaian Red Anthill Clay, Sawdust, Rice Husk

Arthur and Gikunoo [40] studied the properties of thermal insulation materials made from Ghanaian red anthill clay (RAC, % wt.: $SiO_2$—52.35; $Al_2O_3$—31.16; MgO—0.43; $P_2O_5$—0.11; $SO_4$—0.02; $K_2O$—0.73; $TiO_2$—0.51; MnO—0.02; $Fe_2O_3$—4.49) enhanced with non-traditional additives. After sintering, the chemical composition of RAC remained virtually unchanged, although this could not be said about the phase composition. Initially, the presence of kaolinite, quartz and hematite was recorded. After sintering, the formation of mullite as a result of transformation of kaolinite through the metakaolinite phase and spinel was established. The hematite remained virtually unchanged.

The objective of the study was not only to obtain improved refractories but also to make them environmentally friendly, sustainable and economically attractive. Since, as noted above, the performance properties of insulating materials depend largely on their porosity, the authors used combustible additives. As such, sawdust (SD), rice husk (RH) and their mixture were chosen as ingredients, which are very important to dispose of from the environmental point of view. The composition and properties of the experimental samples compared to the control (without additives) sample are shown in Table 7. The introduction of plant additives had no effect on the mineralogical composition of RAC. This shows that identical phases are formed with their participation. As can be seen from Table 7, the experimental samples had a bulk density below the standard value of 1.98 g cm⁻³ [41]. The values of linear shrinkage meet the requirements (10%) for refractory clays [42]. As the plant additives increased, the thermal conductivity of all groups of samples (containing sawdust, rice husk and their mixture) decreased, but remained within the acceptable limits (0.01–1.1 W (m k)⁻¹) for refractory bricks [43]. The compressive strengths of all the experimental samples held standard values (0.981–6.867 MPa) [44]. The required thermal shock-resistance value (20 cycles [45]) was met by the samples with a plant additive content above 15%. In general, studies of the physical, mechanical and thermal properties of the new refractory bricks have shown the potential for using the above materials as raw materials for refractory production, with the introduction of rice husk effective at improving the insulating properties of refractory clay.

**Table 7.** Composition and Properties of Ghanaian Red Anthill Clay-Based with SD/RH additives. Thermal Insulating Materials. Data from [40].

| Samples | Raw Materials, % wt. | | | Apparent Porosity, % | Apparent Bulk Density, g cm$^{-3}$ | Fired Linear Shrinkage, % | Thermal Shock Resistance at 1200 °C, Cycle | Thermal Conductivity, W (m K)$^{-1}$ | Compressive Strength, MPa |
|---|---|---|---|---|---|---|---|---|---|
| | RAC | SD | RH | | | | | | |
| Control | 100 | 0 | 0 | 24.77 | 1.86 | 1.10 | 1 | 0.55 | 2.80 |
| Anthill-SD5% | 95 | 5 | 0 | 33.0 | 1.42 | 1.13 | 11 | 0.46 | 1.82 |
| Anthill-SD10% | 90 | 10 | 0 | 38.7 | 1.28 | 1.25 | 16 | 0.40 | 1.38 |
| Anthill-SD15% | 85 | 15 | 0 | 42.5 | 1.14 | 2.13 | 21 | 0.35 | 1.1 |
| Anthill-SD20% | 80 | 20 | 0 | 52.95 | 1.07 | 2.63 | 25 | 0.31 | 0.74 |
| Anthill-RH5% | 95 | 0 | 5 | 22.5 | 1.52 | 1.50 | 8 | 0.40 | 2.32 |
| Anthill-RH10% | 90 | 0 | 10 | 32.5 | 1.36 | 1.90 | 13 | 0.33 | 1.80 |
| Anthill-RH15% | 85 | 0 | 15 | 36.5 | 1.2 | 2.63 | 17.5 | 0.30 | 1.50 |
| Anthill-RH20% | 80 | 0 | 20 | 40.0 | 1.23 | 3.58 | 20 | 0.23 | 1.05 |
| Anthill-SD-RH5% | 95 | 2.5 | 2.5 | 31.0 | 1.5 | 1.10 | 10 | 0.44 | 2.09 |
| Anthill-SD-RH10% | 90 | 5 | 5 | 37.5 | 1.3 | 1.30 | 14 | 0.38 | 1.60 |
| Anthill-SD-RH15% | 85 | 7.5 | 7.5 | 41.25 | 1.22 | 1.60 | 19 | 0.32 | 1.12 |
| Anthill-SD-RH20% | 80 | 10 | 10 | 42.5 | 1.15 | 2.10 | 22.5 | 0.25 | 0.9 |

### 2.8. Rice Husk Ash, Waste Sediment from Aluminum Anodizing Process, Dregs

Sanewiruch and Saewong [46] investigated the possibility of producing Ca-Al-Si-O compounds as a basis for insulating refractory materials using mixtures of three types of wastes: rice husk ash (RHA, % wt.: $Na_2O$—0.1; MgO—0.7; $Al_2O_3$—0.73; $SiO_2$—88.8; $P_2O_5$—1.1; $SO_3$—0.4; Cl—0.5; $K_2O$—3.3; CaO—3.6; SrO—trace; $Mn_2O_3$—0.2; $Fe_2O_3$—0.4; BaO—trace), waste sediment from the aluminum anodizing process (WS, % wt.: $Na_2O$—4.4; MgO—2.8; $Al_2O_3$—trace; $Al(OH)_3$—79.8; $SiO_2$—1.2; $P_2O_5$—0.1; $SO_3$—6.2; Cl—trace; $K_2O$—trace; CaO—4.3; SrO—trace; $Mn_2O_3$—trace; $Fe_2O_3$—0.7; BaO—trace) and dregs (% wt.: $Na_2O$—9.9; $Mg(OH)_2$—18.0; $Al_2O_3$—1.0; $SiO_2$—2.9; $P_2O_5$—0.19; $SO_3$—5.8; Cl—0.4; $K_2O$—1.9; $CaCO_3$—56.0; SrO—0.1; $Mn_2O_3$—2.7; $Fe_2O_3$—0.9; BaO—0.1).

The compositions of the tested composites are presented in Table 8. According to the XRD data, the main phase of RHA is cristobalite, combined with amorphous silica and a small amount of quartz. WS mainly contains $Al(OH)_3$ and dregs contain $CaCO_3$ and $Mg(OH)_2$. The samples were wet milled and slip-cast in plaster molds to form rectangular bars. Then, the samples were dried and sintered at 1100 °C for 1 h. After sintering, the mineralogical compositions of samples Mix 1, Mix 2 and Mix 3 differed significantly from each other. In Mix 1, the presence of gelenite, mervinite and spinel was noted. Meanwhile, gelenite was absent from Mix 2. Anorthite was detected as the main phase, which was present along with spinel, residuals of cristobalite and aluminum oxide. In Mix 3, gelenite, anorthite, spinel and wollastonite were present with some residual aluminum oxide. According to the study of the microstructure, Mix 1 was formed by needle-like particles of predominantly gelenite. Mix 2, on the contrary, was represented by polygonal grains of <1 μm. Then, in Mix 3, the presence of both particles was observed.

**Table 8.** Composition and properties of Ca-Al-Si-O-based insulating refractory materials. Data from [46].

| Samples | Composition, % wt. | | | Density, g cm$^{-3}$ | Water Absorption, % | Coefficient of Thermal Expansion ($\times 10^{-6}$ C$^{-1}$) | Flexural Strength, MPa |
|---|---|---|---|---|---|---|---|
| | SiO$_2$ | Al$_2$O$_3$ | CaO | | | | |
| Mix 1 | 20 | 20 | 34 | 1.42 | 37.40 | 10.1 | 9.56 ± 0.86 |
| Mix 2 | 49 | 28 | 13 | 0.95 | 67.95 | 6.14 | 4.09 ± 0.99 |
| Mix 3 | 22 | 43 | 19 | 1.17 | 51.13 | 8.03 | 6.28 ± 0.73 |
| Insulating Fire Brick [45] * | - | - | - | 0.5–0.6 | - | 5–6 | 0.7–1.4 |

* [47], cited from [46].

The properties of the obtained composites, presented in Table 8 in comparison to the characteristics of insulating refractory bricks, clearly demonstrate the good prospects for using the mentioned wastes as precursors in refractory production, if the ratio is correct and the conditions of the production process are suitable.

*2.9. Porous Silica of Rice Husk Ash*

Ahmed et al. [48] investigated silica samples of rice husk ash (% wt.: SiO$_2$—66.3; Al$_2$O$_3$—0.064; Fe$_2$O$_3$—0.78; CaO—1.92; K$_2$O—6.2; P$_2$O$_5$—0.07; loss of ignition—24.08%), obtained at different pressures of compaction (P), firing temperatures (T) and soaking times ($\tau$, Table 9) for use in ladle lining in steel production. Since the most important properties of the materials used for ladle lining are porosity and compressive strength, they studied the pattern of change in these characteristics depending on the conditions under which the experimental samples were obtained. It was found that the porosity of rice husk ash silica compacts decreases with an increasing firing temperature and pressure at which they are pressed. The firing temperature is the determining parameter for changes in the degree of porosity. Although an increase in the firing temperature of the samples should increase the porosity as a result of carbon gasification, the degree of porosity in this case, on the contrary, decreases as a result of the high sinterability of the material. An increase in the pressure at all firing temperatures has a favorable effect on the compressive strength of compacts, while changing the firing time has an ambiguous effect on this. At low firing times, an increase in temperature up to 1125 °C causes a slight decrease in compressive strength. With a further rise in temperature, the compressive strength increases again. This proves that, in addition to the porosity and carbon content in the charge, the compressive strength of finished samples is influenced by polymorphic transformations of quartz-cristobalite-tridymite silica, accompanied by changes in the specific density and specific volume. The growth of silica crystals in the transition from one phase state to another causes an increase in volume and a decrease in strength. In concluding the study, optimal modes of production of porous silica compacts (compaction pressure—7.5–14.9 MPa, firing temperature—1000–1250 °C, soaking time—120 min), providing the required indicators of porosity (30%) and compressive strength (2.5 MPa) for use in lining ladles, were recommended.

**Table 9.** Production condition and properties of rice husk ash silica compacts. Data from [48].

| Samples | Production Condition | | | Properties | |
|---|---|---|---|---|---|
| | T, °C | $\tau$, Min | P, MPa | Total Porosity, % | Compressive Strength, MPa |
| 1 | 1000 | 15 | 112 | 70.77 | 6.69 |
| 2 | 1000 | 120 | 75 | 72.00 | 4.69 |
| 3 | 1000 | 120 | 149 | 69.79 | 8.53 |
| 4 | 1000 | 225 | 112 | 69.70 | 6.10 |

**Table 9.** *Cont.*

| Samples | Production Condition | | | Properties | |
|---|---|---|---|---|---|
| | T, °C | τ, Min | P, MPa | Total Porosity, % | Compressive Strength, MPa |
| 5 | 1125 | 15 | 75 | 71.90 | 5.02 |
| 6 | 1125 | 15 | 149 | 68.50 | 8.59 |
| 7 | 1125 | 120 | 112 | 69.79 | 5.80 |
| 8 | 1125 | 120 | 112 | 69.80 | 5.90 |
| 9 | 1125 | 120 | 112 | 69.93 | 5.60 |
| 10 | 1125 | 225 | 75 | 70.20 | 6.43 |
| 11 | 1125 | 225 | 149 | 68.30 | 9.36 |
| 12 | 1250 | 15 | 112 | 69.30 | 5.85 |
| 13 | 1250 | 120 | 75 | 68.98 | 6.44 |
| 14 | 1250 | 120 | 149 | 65.77 | 9.53 |
| 15 | 1250 | 225 | 112 | 69.00 | 7.02 |

## 3. High-Strength Refractory

### 3.1. Quartz, Rice Husk Ash, Clay, Refractory Grog

Bhardwaj et al. [49] investigated the possibility of replacing quartz with rice husk ash used in a mixture with clay and refractory grog (Table 10), to produce high-strength refractory materials. RHA was used from rice mills. In these mills, rice husk is usually used as fuel. After undergoing combustion at 500 °C, RHA is practically free of any volatile substances, such as carbon. However, the RHA was treated at 600 °C for 2 h to remove any residual carbon content. Using an X-ray fluorescence spectrometer, the following composition of RHA, % wt. was established: $SiO_2$—92.81; $Na_2O$—2.658; $P_2O_5$—1.071; $K_2O$—1.021; CaO—0.417; $Fe_2O_3$—0.312; MgO—0.212; $RuO_2$—0.151; $SO_3$—0.132; $TiO_2$—0.112; ZnO—0.091; CuO—0.058; $Rb_2O$—0.036; BaO—0.031; $ZrO_2$—0.025; $Re_2O_7$—0.021; $Y_2O_3$—0.012; $Eu_2O_3$—0.010. The SEM method showed that the outer surface of RHA particles has a pronounced relief. The inner surface is porous, which explains the high value of the specific surface area. According to XRD data, the diffractogram of RHA has no sharp peaks. The researchers recorded only one broad halo in the region of 2θ = 20–30°. This character of the diffractogram indicated the absence of any crystalline phases in the studied sample. Amorphous silica is reactive due to the absence of long-range ordering. When interacting with other materials, it provides higher compaction. The C composition is mainly formed by silicon dioxide (52.8% wt.) and aluminum oxide (33.74% wt.). In small quantities' fixed presence, % wt.: $Fe_2O_3$—0.41; $TiO_2$—0.06; MgO—0.66; $Na_2O$—1.28; $K_2O$—0.7.

To prepare the refractory samples, quartz (90 μm), RHA (106 μm), C (150 μm) and refractory grog (1000 μm) were mixed in an agate mortar using a pestle in the quantities shown in Table 10. Then, the necessary amount of water was added for shaping. The samples were pressed on a uniaxial press at 120 MPa at 40 × 10 mm and air-dried in an electric oven at 110 °C for 24 h. Then, they were heated in a muffle from room temperature to 500 °C for 4 h and then to 1200 °C for 5 h. Sintering was performed for 2 h, after which the samples were cooled to room temperature for 8 h.

In general, the results showed that the experimental refractory samples (Table 10) were characterized by good physical, chemical and thermal properties. Increasing the amount of rice husk ash in the refractory composition, when introduced instead of quartz, contributed to the compaction and hardening of the obtained materials. This is explained by the fact that the amorphous silica of the rice husk ash exceeds quartz in reactivity and more actively interacts with other ingredients during sintering of the charge. When reducing the porosity of refractories with increasing RHA in their composition, increased sample

shrinkage and increased thermal conductivity are observed. Samples with a higher RHA content show a lower refractoriness (PCE). The authors of the considered work explain this fact by the lower melting temperature of the rice husk ash silica (1440 °C) compared to quartz (although in [48], the melting temperature of 1600 °C is indicated as a significant advantage of the rice husk silica). The quartz-based refractory showed a maximum PCE temperature of 1580 °C and the RHA-based refractory showed one of 1470 °C. Sample 20Q30R, the production of which involves the replacement of quartz with ash from rice husk at 30% wt., was the most suitable material for furnace lining.

**Table 10.** Composition and properties of quartz/RHA/clay/grog refractories. Data from [49].

| Samples | Raw Materials, % wt. | | | | Apparent Porosity, % | Bulk Density, gm cc | Linear Shrinkage, % | Cold Crushing Strength, MPa | Thermal Conductivity, W (m K)$^{-1}$ | | | Pyrometric Cone Equivalent (PCE) |
|---|---|---|---|---|---|---|---|---|---|---|---|---|
| | Quartz | RHA | C | Grog | | | | | 200 °C | 600 °C | 800 °C | |
| 50Q | 50 | 0 | 40 | 10 | 21 | 1.72 | 3.9 | 30.6 | 1.12 | 1.46 | 1.9 | 20 |
| 40Q10R | 40 | 10 | 40 | 10 | 17.8 | 1.81 | 4.1 | 34 | 1.22 | 1.58 | 1.91 | 20 |
| 30Q20R | 30 | 20 | 40 | 10 | 15.4 | 1.90 | 4.75 | 36.1 | 1.3 | 1.64 | 1.98 | 19 |
| 20Q30R | 20 | 30 | 40 | 10 | 14.6 | 1.94 | 5.2 | 38.0 | 1.31 | 1.7 | 2.08 | 18 |
| 10Q40R | 10 | 40 | 40 | 10 | 13.8 | 1.97 | 5.65 | 38.4 | 1.38 | 1.72 | 2.12 | 17 |
| 50R | 0 | 50 | 40 | 10 | 13.2 | 2.00 | 6.1 | 38.3 | 1.38 | 1.82 | 2.22 | 16 |

According to XRD data, the intensity of crystallinity of this sample increased significantly compared to sample 50Q. The crystallization process began at 800 °C. At 1000 °C, the sample had a completely crystalline structure represented by cristobalite, tridymite and mullite. The surface morphology had also changed. The distinctive cut of the particles characteristic of the 50Q sample practically disappeared in the 20Q30R sample after sintering at 1200 °C. The surface became smoother as a result of the transformation of amorphous silicon dioxide into a crystalline state.

In general, the promising characteristics of this sample indicate the real prospect of creating high-strength clay-based refractories with rice husk ash for use in most furnace linings where silica refractories are commonly used.

### 3.2. Rice Husk Silica, Al$_2$O$_3$, MgO

One of the excellent refractory materials characterized by the highest melting point among silicate ceramics, high chemical stability and an excellent thermal shock resistance is cordierite. Sembiring et al. [50,51] investigated the possibility of producing this material from rice husk silica at different sintering temperatures [50] and charge ingredient ratios [51]. Silica was obtained by treatment of rice husk with alkali under boiling followed by hydrochloric acid precipitation. Powders of magnesium, aluminum and silicon oxides were mixed in the ratio of 2:2:5 by mass, respectively. Then, these ingredients were mixed with alcohol in a magnetic stirrer for 6 h. After the mixing process, the mixture was filtered off. The solid was dried at 110 °C for 8 h to remove the residual alcohol. The dried solid was ground in a mortar and sifted through a 200-mesh sieve. The powder was pressed in a metal mold at a pressure of $2 \times 10^4$ N m$^{-2}$ to obtain cylindrical pellets. The pellets were

sintered at temperatures of 1050, 1110, 1170, 1230, 1290 and 1350 °C for 4 h by heating at 3 °C min$^{-1}$.

The phase changes with an increasing sintering temperature were studied by FTIR, XRD and SEM methods. At temperatures above 1110 °C, using FTIR, the appearance of new bands at 640, 15, 590, 460 and 430 cm$^{-1}$ was observed. Moreover, the higher the sintering temperature, the higher the intensity of these bands, confirming the presence of MgO–Al$_2$O$_3$–SiO$_2$ structures. According to XRD data, the presence of cristobalite, μ-cordierite and spinel was detected in the sample obtained at 1050 °C. The formation of cristobalite is due to crystallization of silicon dioxide of rice husk. The presence of μ-cordierite is the result of intra-diffusive processes of interaction between spinel and cristobalite. At 1110 °C, μ-cordierite transforms into α-cordierite. The presence of these phases remains up to 1170 °C. It was found that the α-cordierite phase was predominant at a sintering temperature of 1230 °C. At this time, the reflexes of spinel and cristobalite disappear almost completely. In the region of 1230–1350 °C, cristobalite melts and reacts with the spinel phase, forming α-cordierite. Its amount at 1350 °C reaches 92.7% wt., against 30.8% wt. at 1110 °C. The surface morphology with an increasing sintering temperature changes from small, clearly faceted grains at 1050 °C to large, agglomerated particles with a glassy surface, without evidence of grain boundaries, at 1230–1350 °C.

The sample obtained at 1230 °C was characterized by hardness and high bending strength, while the porosity, density and thermal expansion coefficient corresponded to the formation of the cordierite phase (Table 11). Aluminum-cordierite samples with different mass ratios of cordierite to aluminum oxide (100:0, 95:5, 90:10, 85:15, 80:20, 75:25, 70:30) were obtained from this material. As can be seen from Table 12, aluminum-enriched refractory cordierite is mainly composed of spinel, corundum and cristobalite. Increased aluminum suppresses the growth of cordierite crystals. The XRD data of a Rietveld analysis (Table 12) demonstrate that a binary interaction between MgO and Al$_2$O$_3$ producing spinel is preferable to the interaction between MgO and SiO$_2$. Thus, the spinel formation process is the result of an intra-diffusive interaction of aluminum oxide with periclase. Significant changes in the surface morphology of samples with increasing aluminum oxide addition were recorded by SEM. As a result of aluminum addition, fine grains of α-cordierite were destroyed, with the formation of agglomerated spinel particles, corundum and cristobalite.

**Table 11.** Composition and properties of refractory cordierite from rice husk silica. Data from [50].

| Temperature, °C | Phase Composition, % wt. | | | | Porosity, % | Density, g cm$^{-3}$ | Bending Strength, MPa | Hardness, GPa | Coefficient of Thermal Expansion, ($\times 10^{-6}$) (°C)$^{-1}$ |
| | α-Cordierite | Cristobalite | Spinel | μ-Cordierite | | | | | |
|---|---|---|---|---|---|---|---|---|---|
| 1050 | - | 39.7 | 32.9 | 28.4 | 23 | 3.5 | 300 | 180 | 9.5 |
| 1110 | 30.8 | 40.1 | 29.1 | - | 15.5 | 3.45 | 350 | 300 | 9.0 |
| 1170 | 33.8 | 39.6 | 26.6 | - | 11 | 3.3 | 450 | 340 | 8.5 |
| 1230 | 90.5 | 4.7 | 4.8 | - | 3 | 2.2 | 830 | 440 | 3.5 |
| 1290 | 91.8 | 3.9 | 4.3 | - | 3.5 | 2.25 | 850 | 480 | 3.4 |
| 1350 | 92.7 | 3.8 | 4.2 | - | 3 | 2.3 | 860 | 520 | 3.3 |

**Table 12.** Composition and properties of $Al_2O_3$-refractory cordierite from rice husk silica. Data from [51].

| $Al_2O_3$, % | Phase Composition, % wt. | | | | | Porosity, % | Density, g cm$^{-3}$ | Bending Strength, MPa | Hardness, GPa | Coefficient of Thermal Expansion, $(\times 10^{-6})$ $(°C)^{-1}$ |
|---|---|---|---|---|---|---|---|---|---|---|
| | $\alpha$-Cordierite | Spinel | Corundum | Cristobalite | Periclase | | | | | |
| 0 | 90.5 | 4.7 | 4.8 | - | - | 26.75 | 2.34 | 65 | 5.8 | 2.46 |
| 5 | 50.6 | 4.1 | 15.2 | 30.1 | - | 25.65 | 2.52 | 120 | 6.0 | 2.52 |
| 10 | 0.9 | 14.5 | 40.2 | 42.3 | 2.1 | 24.59 | 2.60 | 155 | 8.0 | 4.0 |
| 15 | 0.6 | 30.8 | 36.9 | 29.4 | 2.3 | 20.0 | 3.51 | 144 | 23.8 | 8.5 |
| 20 | 0.3 | 36.6 | 34.4 | 25.3 | 3.3 | 5.78 | 3.53 | 145 | 23.2 | 9.0 |
| 25 | 0.4 | 41.7 | 31.1 | 24.1 | 2.7 | 4.5 | 3.70 | 144 | 23.1 | 9.2 |
| 30 | 0.4 | 45.4 | 28.7 | 22.5 | 3.0 | 3.0 | 3.72 | 140 | 23.0 | 9.5 |

Aluminum-enriched refractory samples have high values of density, hardness, flexural strength and thermal expansion coefficient and, on the contrary, a low porosity index. Rice husk silica-based refractory cordierite samples represent a promising insulating material, and aluminum-enriched, dense samples can be used as abrasive materials.

### 3.3. Rice Husk Ash, $\alpha$-$Al_2O_3$

In [52], ash from the industrial combustion of rice husk was used as a source of $SiO_2$ to produce mullite ceramics. The chemical composition of RHA was as follows % wt.: $SiO_2$—94.72; $SO_3$—0.17; $CO_2$—0.23; $P_2O_5$—0.51; CaO—0.21; MgO—0.18; $Na_2O$—0.05; $K_2O$—0.46; $Al_2O_3$—<0.05; $Fe_2O_3$—0.05; $Cl^-$—0.02; C—3.39. RHA had a melting point > 1200 °C; calorific power—351 cal g$^{-1}$; specific surface (BET)—11.22 m$^2$ g$^{-1}$. According to XRD, RHA contained cristobalite—67.7% wt.; amorphous—30.2% wt.; orthorhombic Si-O—3.1% wt. The particle size distribution of RHA was as follows % wt.: 1.00 mm—0.78; 0.595 mm—5.61; 0.297 mm—42.05; 0.177 mm—24.32; 0.149 mm—5.43; 0.125 mm—6.81; 0.105 mm—2.33; ≤0.105 mm—12.62.

Only $\alpha$-$Al_2O_3$ in a stoichiometric ratio of $3Al_2O_3$:$2SiO_2$ was used as the other ingredient. The ingredients were milled in a ball mill (1000 cm$^3$ for 24 h) and dry-pressed with a pressure of 200 MPa. Disc-shaped samples ($\varnothing$ = 15.00 mm) were fired in an electric kiln at temperatures from 1100 to 1600 °C for 60 min with a heating rate of 5 °C min$^{-1}$. The phase composition and properties of the obtained materials are shown in Table 13. After comparing the results of a number of analyses (XRD, SEM) and a study of the properties of the created samples, the authors concluded that the mullitization reaction begins at 1400 °C and ends at 1600 °C. The lack of $SiO_2$ is explained by its interaction with inorganic impurities present in rice husk ash. A decrease in the density of the obtained products relative to the density of the initial components, and sample shrinkage at 1400 °C, in their opinion, confirm mullite's formation. The structure and properties of the obtained ceramic materials open up the possibility of their wide use. For example, it is possible to increase the porosity of the final product by introducing combustible additives for its use as insulating materials. On the contrary, to obtain denser materials, strengthening the grinding of the initial components in the preparation stage is suggested. [52] provided enough information to control the process and thereby obtain mullite ceramics with specified properties using rice husk ash.

**Table 13.** Composition and properties of mullite ($3Al_2O_3 \cdot 2SiO_2$) ceramics produced using rice husk ash. Data from [52].

| Temperature, °C | Phase Composition, % wt. | | | Open Porosity, % | Apparent Density, g cm$^{-3}$ | Shrinkage, % |
|---|---|---|---|---|---|---|
| | Corundum | Cristobalite | Mullite | | | |
| 1100 | 88 | 12 | - | 45 | 3.4 | 1 |
| 1200 | 85 | 15 | - | 40 | 3.3 | 2 |
| 1300 | 90 | 10 | - | 35 | 3.2 | 3 |
| 1400 | 87 | 5 | 8 | 33 | 3.3 | 8 |
| 1500 | 30 | - | 70 | 32 | 3.2 | 6 |
| 1600 | 8 | - | 92 | 34 | 3.2 | 5 |

*3.4. Rice Husk Ash, Quartz, MgO*

In [53], the possibility of obtaining another type of durable refractory ceramic was shown. We are talking about forsterite refractory, which was prepared using quartz (98% pure) and periclase (99% pure), gradually replacing the quartz with rice husk ash up to complete substitution (Table 14). The characteristics of the RHA were published elsewhere [49].

**Table 14.** Composition and properties of forsterite ceramics produced using rice husk ash. Data from [53].

| Samples | Composition, % wt. | | | Apparent Porosity, % | Bulk Density, gm cc | Cold Crushing Strength, MPa | Thermal Conductivity, W (m K)$^{-1}$ | | |
|---|---|---|---|---|---|---|---|---|---|
| | MgO | Quartz | RHA | | | | 200 °C | 600 °C | 1000 °C |
| 1 | 57 | 33 | 10 | 23.17 | 2.14 | 11.76 | 2.576 | 3.695 | 4.173 |
| 2 | 57 | 23 | 20 | 19.00 | 2.23 | 13.38 | 2.145 | 2.732 | 3.78 |
| 3 | 57 | 13 | 30 | 14.65 | 2.32 | 15.2 | 1.835 | 2.289 | 3.192 |
| 4 | 57 | 00 | 43 | 10.54 | 2.40 | 18.63 | 1.039 | 1.357 | 2.278 |

To prepare refractory samples, all ingredients (quartz—90 μm, RHA—106 μm, MgO—106 μm) were mixed in two steps. Dry mixing was carried out for 30 min and semi-dry mixing (with the addition of water as a medium) for 20 min. The samples were pressed using a hydraulic press at 123 MPa. Sintering was carried out at 1100 °C for 2 h at a heating and cooling rate of 5 °C min$^{-1}$.

The formation of the forsterite phase was proven using XRD by the presence of the following bands at 2θ = 20.6°, 41.80° and 62.008°. The characteristic peaks of forsterite in the fired samples were identified by FTIR analysis, for example, 500–620 cm$^{-1}$ (octahedral $MgO_6$ or $SiO_4$ bending modes), 650–840 cm$^{-1}$ (Si-O-Si symmetric stretching vibrations), 830–1000 cm$^{-1}$ (stretching vibrations of nonbridging Si-OH) and 1000–1050 cm$^{-1}$ (Si-O-Si asymmetric stretching vibrations). Based on EDX data, amorphous silica extracted from rice husk during its combustion was found to react much more actively with periclase at 1100 °C than with quartz. RHA additives affect the surface morphology of particles and their size distribution, as noted by SEM. The higher the RHA content, the stronger the forsterite formation and the denser the material that is formed. With an increase in the amount of rice husk in the charge of the obtained refractory, a reduction in porosity, a decrease in thermal conductivity, along with increases in the cold crushing strength and density of the finished samples, were observed (Table 14). However, the complete substitution of quartz by rice husk ash resulted in a material with a density (2.4 gm cc) lower than the theoretical one (3.2–3.3 gm cc) for forsterite, which was explained by the predominance of a closed porosity, which reduced the thermal conductivity. Considering

the high density and low thermal conductivity of rice husk ash-based forsterite refractory, this material is recommended for use as thermal insulation in aggregates for steel and cement production.

## 4. SiC-Based Refractory Compounds

Important ceramic materials for industrial applications, especially at high temperatures, are silicon carbide and products based on it. Silicon carbide combines a set of excellent mechanical, physicochemical, thermal and electrical properties. It has hardness, high strength, thermal conductivity, is characterized by low thermal expansion, is chemically inert, stable in oxidizing environments and is not subject to erosion or corrosion. There are $\alpha$-SiC and $\beta$-SiC polytypes of silicon carbide that differ in their structure, morphology and properties and are formed under different conditions. There are multistage and single-stage processes of silicon carbide production. Its modern production requires the use of simple techniques and cheap raw materials [54,55]. For the latter, the use of ground agricultural waste, such as wheat hulls, corn cobs, sorghum leaves, peanut peels and others, is suggested [56–61]. Rice husk, as well as its ash, is a very attractive raw material source in this regard due to its high reactivity and purity, high content of carbon and silicon and their close interaction [19,62–68]. Moreover, of interest is the carbothermic production method [69,70], which is more cost-effective compared to chemical vapor deposition and the sol-gel method.

Analysis of the results of studies of the process of thermal degradation of rice husk obtained by different authors [16,71–85] shows that the product of rice husk decomposition in the absence of oxygen (in an inert environment, a vacuum, in a waste gas atmosphere) is a silica-carbon nanocomposite (black ash, char, biochar) formed by carbon and silicon dioxide nanoparticles. The structure and morphology of the particles depend on the conditions of the destruction process. Carbon and silicon dioxide are present in an amorphous form [66]. However, carbon has a graphite-like structure [75,79,86]. As the carbonization temperature rises, the degree of ordering of the graphite-like structure increases. As for the silicon-containing phase, up to 800 °C, silicon dioxide is mainly in the amorphous form [74,77]. The presence of $H_2Si_{14}O_{29} \cdot 5,4H_2O$ was registered during pyrolysis of rice husk in an exhaust gas atmosphere at 650 °C [78], although [72] described that $\alpha$-quartz formation was observed in a nitrogen atmosphere at temperatures below 800 °C. Cristobalite appears at temperatures above 800 °C in different atmospheres [71,72,75,77]. In a nitrogen atmosphere at 900 °C, rice husk modified with sodium hydroxide solution decomposes to form graphitic carbon, cristobalite and tridymite [87]. Researchers note that the process of crystallization proceeds most actively in the air [71]. The presence of different silica phases up to tridymite is noted in the product of rice husk combustion in the air [78]. Javed et al. [88] found that pretreatment of rice husk with a dilute solution of potassium permanganate during pyrolysis in the temperature range of 500–700 °C promotes faster decomposition of the organic component but inhibits the crystallization of silica. Silicon carbide formation occurs during the carbonization of rice husk or a mixture of its ash and carbon at higher temperatures (1300–1600 °C and above) [62,63,66–68,77,89–91]. Despite the known fact that silicon carbide is used as an enhancer of metal and ceramic composites, and though there are examples of silicon carbide being obtained from rice husk, there is little information in the literature about how to obtain refractories based on silicon carbide from rice husk.

### Al-Mg-Si Alloy-Based Composites

The authors of [32] investigated the microstructural characteristics and mechanical properties of aluminum matrix composites reinforced with 10% silicon carbide (SRC) from rice husk ash (RHA, % wt.: $SiO_2$—91.81; C—4.91; CaO—1.35; MgO—0.50; $K_2O$—0.41; $Fe_2O_3$—0.29; others—0.73). The latter were obtained by the carbothermic method (Table 15). Al-Mg-Si alloy-based composites (% wt.: Mg—0.35; Si—0.59; Mn—0.35; Cu—0.012; Zn—0.002; Ti—0.057; Fe—0.47; Ni—0.035; Al—balance) were prepared by a double-stir casting process. SRC was preheated at 200–280 °C. The Al-Mg-Si alloys were heated in a

furnace to a temperature of $710 \pm 30$ °C until they completely melted. Then, they were cooled to a semi-solid state and heated again to a temperature of $710 \pm 30$ °C. Finally, they were stirred at 350 rpm for 5–10 min before casting in a sand mold.

**Table 15.** Characteristics of Al-Mg-Si alloy-based composites. Data from [32].

| Samples | Conditions of SRC Production | Theoretical Density, g cm$^{-3}$ | Experimental Density, g cm$^{-3}$ | Porosity, % | Hardness, HV0.1 | Ultimate Tensile Strength, MPa | Yield Strength, MPa | Strain to Fracture, % | Fracture Toughness, MPa m$^{1\cdot(-2)}$ |
|---|---|---|---|---|---|---|---|---|---|
| A1650 | at 1650 °C without catalyst | 2.649 | 2.565 | 3.171 | 76 | 120 | 98 | 19 | 4.4 |
| C1650 | at 1650 °C in catalytic environment | 2.649 | 2.595 | 2.039 | 79 | 140 | 108 | 23 | 5.2 |
| A1600 | at 1600 °C without catalyst | 2.649 | 2.559 | 3.398 | 70 | 102 | 90 | 14 | 3.3 |
| A1250 | at 1250 °C without catalyst | 2.649 | 2.55 | 3.737 | 59 | 84 | 82 | 15.5 | 3.8 |
| B1250 | at 1250 °C with the initial powder unconditioned and later treated in catalytic environment | 2.649 | 2.549 | 3.775 | 56 | 90 | 60 | 19 | 4 |
| C1250 | at 1250 °C with the initial powder conditioned in catalytic environment | 2.649 | 2.546 | 3.888 | 62 | 92 | 64 | 20 | 3.7 |
| D1 | conventional SiC | 2.743 | 2.690 | 1.9 | 82 | 160 | 122 | 15 | 4.4 |

In studying the composition of the refractory material made of rice husk, the presence of SiC polytypes as the main reinforcing component was discovered. In addition, intermetallides were found in the composition of the obtained composites, certainly contributing to their strength [32]. Samples with higher silicon carbide contents were less porous, harder and had higher ultimate tensile strength and yield strength values. However, their crack resistance values were not high because silicon carbide is a hard reinforcing material that causes rapid crack propagation.

## 5. Conclusions

Summarizing the review, the following conclusions can be drawn:

- Rice husk ash is an alternative renewable source of silica for the production of refractory materials that meet the set requirements. It can be used as an independent ingredient of the charge for their production or as a substitute for some raw materials traditionally used for this purpose;
- Rice husk can be used to create refractories not only as a silica raw material but also simultaneously as a combustible additive (due to the presence of an organic component), as well as carbon-silica raw materials, and for the synthesis of silicon carbide;
- The presence of silica in rice husk/rice husk ash in the amorphous form ensures its high reactivity when interacting with other ingredients used in the refractory manufacturing process. This makes it possible to obtain the desired products at lower temperatures and with less time spent;
- Application of rice husk/rice husk ash in the process under consideration makes it possible to create different types of refractory materials, e.g., porous, high-strength or made of silicon carbide. The formation of their physical-mechanical and thermal properties is determined and, accordingly, controlled by the selection of the optimal composition and the necessary synthesis of operating conditions;

- The application of rice husk ash is equally successful for the production of both porous and high-strength refractories. The introduction of rice husk ash into their composition improves their mechanical and thermal properties. However, the mechanism of action of this ingredient has not yet been fully discovered. In this regard, further studies are required to determine whether it is really a universal raw material and how it improves certain physical and mechanical properties;
- There is no discussion in the literature about the influence of impure elements present in rice husk ash on $SiO_2$ crystallization processes and, correspondingly, the formation of the main physical and chemical properties of refractory materials. Elimination of this research gap seems necessary in future studies;
- Rice husk silica-based refractory materials have wide uses. They are promising thermal insulators for different kinds of furnaces, lining ladles, as aggregates for steel and cement production and as abrasive materials. However, the available works on the application of rice husk ash for the production of refractories are of a purely scientific nature. There is no analysis of the economic efficiency of replacing traditional raw materials with alternative materials. There are no examples of industrial production of refractory materials or at least industrial tests of their production with the use of rice husk ash. The idea of recycling rice husk ash for refractory production in the future should be confirmed on an industrial scale;
- The results of studies in the field of rice husk/rice husk ash to create various refractory materials are prerequisites for the birth of an independent research area. Its task is to resolve the outstanding issues and build a solid foundation for the organization of industrial production, with a real contribution to the development of the refractory industry and related industries founded on the principles of environmental safety.

**Author Contributions:** Conceptualization, S.Y. (Svetlana Yefremova) and B.S.; methodology, S.Y. (Svetlana Yefremova); investigation, S.Y. (Svetlana Yefremova), B.S., N.S., S.S., S.Y. (Sergey Yermishin) and A.K.; data curation, S.Y. (Svetlana Yefremova); writing—original draft preparation, S.Y. (Svetlana Yefremova); writing—review and editing, A.Z.; supervision, A.Z. and B.S.; project administration, S.Y. (Svetlana Yefremova); funding acquisition, A.Z. and B.S. All authors have read and agreed to the published version of the manuscript.

**Funding:** This research was carried out as a part of a project initiative, supported by National Center on Complex Processing of Mineral Raw Materials of the Republic of Kazakhstan.

**Acknowledgments:** The authors would like to thank the journal editors for their invitation to submit a manuscript and the reviewers for their revision of this paper.

**Conflicts of Interest:** The authors declare no conflict of interest.

## Abbreviations

The following abbreviations are used in the manuscript:

| | |
|---|---|
| B | bentonite |
| C | clay |
| DE | diatomite |
| DS | industrial diatomaceous silica |
| Ex | extrusion method |
| FTIR | Fourier-transform infrared spectroscopy |
| KC | kaolin |
| Pr | pressing method |
| PS1 | polysaccharide |
| PVA | polyvinyl alcohol |
| RAC | red anthill clay |
| RH | rice husk |
| RHA | rice husk ash |

RHS        rice husk silica
SD         sawdust
SEM        scanning electron microscopy
SF         steel fibers
SHMP       sodium hexametaphosphate
SRC        silicon carbide
W          wollastonite
WS         waste sediment
XRD        X-ray diffraction

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
