# Peer review of "Production of Refractory Materials Using a Renewable Source of Silicon Dioxide"

_minerals, doi:10.3390/min12081010_

Round 1
Reviewer 1 Report
The review is important, it related to current environmental and ceramic industry’s needs. This review was of fair quality in terms of novelty and significance. I recommended it to be published in ‘Minerals’ after minor revision. The following points should be addressed:
Please do a thorough review and include a survey of the current literature available on this topic. A deep and up-to-date review is needed.
Some discussion about rice husk silica is need in the review, please include.
Please include an abbreviation list in the manuscript.
It is suggested to add some figures and photographs to the manuscript.
One of sections is needed to present the LCA analysis on environmental impact and cost analysis compared the commercial silica and RHA silica used refractory.
Please include some discussion about potential applications of RHA refractory.
Objective of the review is not clear in Introduction. Need to rewrite. English is poor, needing strong improvement.
Please include prospect of the review.
In the conclusion, please show how this review advances the field from the present state of knowledge. Please provide a clear justification for this review in this section.
Author Response
Dear Reviewer,
Authors would like to thank you for your time spent reviewing the manuscript and useful recommendations.
Your comments are very valuable for us. In general, we agree with your point of view. Therefore, we paid attention to each your suggestion.
In our opinion, revision done according to your comments really helped to improve the manuscript.
Please see our response attached.
Thank you again.
Best regards,
Prof. Svetlana Yefremova

Reviewer 2 Report
The paper describes the synthesis of refractory materials using rice husk ash and other silicon-containing wastes such as diatomite, sawdust, refractory clay, and wollastonite microfibers, etc. The physical properties of the synthesized materials including engineering, mechanical and thermal properties were investigated. The topic is interesting. However, there are a number of negative aspects of the work, of which I mention just a few of the more obvious flaws:
1. Previously published literature have established the probability of using rice husk ash into refractory materials. The novelty character of paper should be marked.
2. In introduction, it is advisable to include the recent papers on the basic characteristics of rice husk ash such as the production, physical and chemical characteristics, microstructure, and applications of RHA.
3. In order to avoid confusing, the abbreviation in the paper should be identical. For example, rice husk ash is RHA and sawdust is SD.
4. On page 3, Line 2. What is the definition of 90 um and 250 um?
5. It requires some more critical data such as SEM and XRD tests, due to data is not enough.
6. What is the advantages and disadvantages for using rice husk ash as a refractory material when compared with other silicon-containing wastes such as diatomite, sawdust, refractory clay, or wollastonite microfibers?
7. In “Section 2.1”. The role of DE, RHA, and SD is not clear, please explain more the effect of DE, RHA, and SD on volumetric weight, porosity, bending strength, compressive strength, thermal shock resistance, and thermal conductivity for refractory materials synthesis.
8. Limits, advantages, practical applications and future directions should be marked in Conclusion.
Author Response
Dear Reviewer,
Authors would like to thank you for your reviewing and comments.
Your recommendations were very useful. We agree with you completely. As a result, we tried to clarify all issues and improve the manuscript.
Please find detail response attached.
Thank you.
Best regards,
Prof. Svetlana Yefremova

Reviewer 3 Report
Dear Authors,
Thank you very much for your manuscript.
The topic is very relevant and the research carried out in recent years by numerous research groups requires a review paper to present the wide range of possible applications.
Some of the relevant applications explored are cited by the authors in this paper. However, essential details on the manufacturing conditions and the characterization of the starting materials and the produced intermediates are missing. These process parameters and properties have a major impact on the qualities of the application and products presented. Thus, no general conclusions can be drawn about the applicability of the materials by this review.
For example, the thermal conversion of biomasses and the properties of the resulting inorganic intermediates depend very strongly on the content of accompanying elements such as potassium and calcium. Potassium accelerates the crystallization of SiO2 and leads already under relatively low temperatures to nonporous and nonreactive silicates, which behave differently in mixtures than amorphous SiO2, which is able to react appropriately with other compounds by solid-state reaction.
Further, without knowing the composition and element contents in the biomasses, it is not possible to obtain reproducible conclusions on application properties of the resulting products.
Another important aspect is the temperatures and residence times during thermal conversion/synthesis, which, together with the composition of the biogenic raw materials, have a major influence on the application properties of the products.
Therefore, I recommend to extend the review with the considered production routes of SiO2 in the products for refractory materials and also with raw material properties and, in particular, to additionally include (if available) the physical and chemical properties of the intermediate products in the reviews. For this reason, I recommend major revisions.
Additional comments on the revision are provided in the attached document.
Best regards
Reviewer

Author Response
Dear Reviewer,
Authors would like to thank you for your deep discussion of the manuscript.
It took a lot of your time to help us to improve the manuscript. It was revised according to your recommendations.
Please see our response attached.
Thank you.
Best regards,
Prof. Svetlana Yefremova

Round 2
Reviewer 2 Report
The paper has been improved regarding the issues noted in the original version. The paper can be recommended for publication.
Reviewer 3 Report
Dear Authors,
Thank you very much for the revised manuscript. I recommend it for publication in this revised version.
With best regards
Reviewer